# Maintaining ecological stability for sustainable economic yields of multispecies fisheries in complex food webs

Alexandra S. Werner[1,2], Myriam R. Hirt [1,2], Remo Ryser[1,2], Kira Lancker[3], Georg Albert [1,2,4], Martin Quaas [3], Christopher Rackauckas[5], Benoit Gauzens [1,2,6] & Ulrich Brose [1,2,6] ✉

Fish stocks are increasingly overexploited due to the growing global demand for seafood. As these species are embedded in complex food webs, traditional single-species management plans should be replaced by models that integrate multi-species fisheries with economic market feedbacks into complex food webs to promote sustainable resource use. Here, we develop such a dynamic model involving three open-access fisheries in a complex food web. Systematically comparing six fishing scenarios, we find that targeting low or high trophic levels risks reducing basal biomass or triggering trophic cascades that undermine first ecological stability (food web biomass and persistence) and then sustainability of economic returns (total sustainable catch and revenue). High sustainable economic returns combined with low negative ecological impacts occur when similar mid-trophic level species are caught in multi-species fisheries. We conclude that complex systems analysis can help design ecosystem-based management strategies to achieve a sustainable food supply for the world.

World food security is an issue of rising importance. With a growing population, we risk increasing ecological damage caused by our food systems, which rely on sustainable natural resource availability[1]. Fish and fish products are one of the most traded goods in the world[2,3], making them particularly important for future food security. Although marine annual fish catch is stable, fishing efforts keep increasing to maintain supply despite reduced fish stocks[3–5], leading to sobering statistics. Current estimates suggest that humans take more marine fish than all natural predators combined[6,7], with 57.3 percent of fished stocks classified as fully exploited (i.e., harvested to their maximum sustainable yield), and 35.4 percent classified as overexploited (i.e., harvested beyond their maximum sustainable yield)[4]. These ecological disruptions could intensify as the growing global human population[1] and increasing demand for fish protein[8,9] incentivize both the expansion of existing and the opening of new fisheries[10]. This raises concerns about impacts on marine biodiversity[11,12] with severe feedback on ecosystem functioning and services[3,13]. The need to keep pace with supplying food and livelihoods while balancing the stability and functioning of ecosystems poses complex sustainability challenges as humans and ecosystems are interlinked[14].

Real-world fisheries are mostly managed using maximum sustainable yields to maximize the value of net revenues, which requires foreseeing fishing impacts on future ecosystem states and fishing opportunities[15]. The economic analysis of multispecies fisheries has largely focused on simple single- or few-species models[16–19] or size-spectrum models describing the community by the distribution of biomass across the body-size classes of species[20,21]. They showed that fishing should target species that turn biological resources most

[1]EcoNetLab, German Centre for Integrative Biodiversity Research (iDiv) Halle-Jena-Leipzig, Leipzig, Germany. [2]Institute of Biodiversity, Friedrich Schiller University Jena, Jena, Germany. [3]Department of Economics, Leipzig University, Leipzig, Germany. [4]Department of Forest Nature Conservation, University of Göttingen, Göttingen, Germany. [5]Department of Mathematics, Massachusetts Institute of Technology, Cambridge, USA. [6]These authors contributed equally: Benoit Gauzens, Ulrich Brose. ✉e-mail: ulrich.brose@idiv.de

efficiently into economic value, either through the highest prices per unit of the biological resource[16,17,21] or through the highest population biomass productivity[20]. Natural ecosystems, however, are characterized by a complexity of food webs that goes beyond simple interaction structures. Intricate linkage patterns create manifold couplings amongst populations causing cascading indirect effects, trophic cascades and feedback loops[22,23]. It is an open question whether the economic scenarios ignoring species' couplings in the network efficiently capture the long-term sustainability of multiple fisheries.

Recent studies have integrated single fisheries in complex food-web models to show that disturbances spread through non-harvested species with severe consequences for fishery strategies[24–26]. Strikingly, targeting high-biomass fish resources can booster short term economic profitability but extinction cascades among non-harvested species threaten the long-term sustainability of this fishery strategy[26]. The strength of these cascading effects varies with the trophic level of the target species[25] as harvesting (i) large species of high top trophic levels can trigger strong trophic cascades[27,28], and (ii) species of low trophic levels, such as herbivorous fish, can cut the energy supply for higher trophic levels[29]. These indirect effects can create strong linkages of fisheries making it essential to integrate multiple fisheries and complex food webs.

To explore the long-term ecological and economic consequences of different fishing strategies, we use a dynamic model coupling simulations of complex food web dynamics with an economic model of multiple fisheries (i.e., multi-fleet fisheries). This way, humans are not considered merely an external disturbance, but are integrated into the food web as three fishery nodes. While predators consume resource biomass according to ecological principles, fishing is guided by short-term economic incentives. Fishing is profitable on species that are the source of preferred seafood for human consumption, whereas other species are priced at zero. Although all species are subject to potential fishing pressure, the fishing fleets select harvestable species based on the economic profitability and market feedbacks. We consider three fishing fleets that each catch one distinct species. The set of species that are selected from the food webs is chosen according to six scenarios including a Random, two economic and three network scenarios (Fig. 1). As outlined above, the two economic scenarios focus on (1) Productivity: harvesting species with the highest biomass productivity to achieve the highest biomass catch of the fleets, and (2) Price: harvesting species with the highest price per gram of weight (Fig. 1A). The interactions amongst fisheries are likely to depend on the harvested species' network positions relative to each other, which can be characterized by distances in terms of trophic

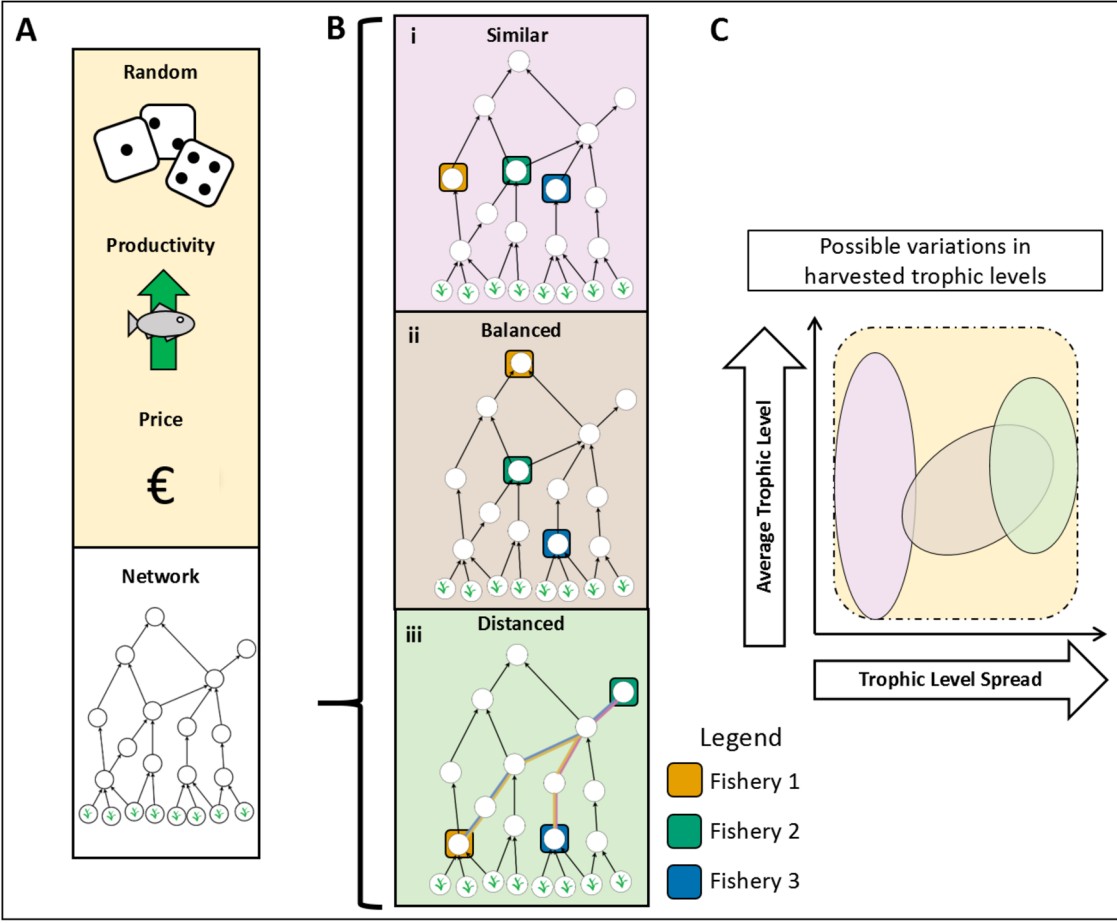

**Fig. 1 | Concept of six fishery strategies. A** The choice of the target species can be (1) Random, or based on economic or network scenarios. The economic scenarios are (2) highest Productivity and (3) Price choosing the three target species with the highest price or highest productivity, respectively. **B** The network scenarios are (4) Similar, where species of similar size and trophic level are fished, (5) Trophically-balanced, where harvested species are chosen to maximize their difference in trophic levels, and (6) Distanced, where species with the largest link distance are harvested. **C** The differences between network scenarios can be characterized by (1) the average and (2) the variation of the trophic levels of the three harvested species. Similar is causing low variation in trophic levels within but high variability in average trophic levels across simulation runs, Trophically-balanced is causing high variation in trophic levels of the harvested species through the food web, and Distanced is causing a high variation of their trophic levels within and variable average in trophic levels across simulation runs. Primary producers or plants (bottom row) are not harvested.

**Table 1 | Definitions of indicator variables used to assess the impacts of our fisheries**

| Indicator | Definition |
|---|---|
| Species persistence | The proportion of species that persisted after the introduction of the fisheries. |
| Food web biomass change (non-harvested) | The difference between the community biomass values at the end of the simulations with fisheries ($B_f$) and at the end of the pristine simulations ($B_p$), divided by the biomass values at the end of the pristine simulations: $(B_f - B_p)/B_p$. Calculated for non-harvested species only. |
| Sustained total biomass catch | The pooled biomass caught of all three fisheries once the system has reached equilibrium, per time step. |
| Sustained total revenue | The pooled revenue of all three fisheries reached once the system has reached equilibrium, per time step. |
| Number of sustained fisheries | The number of fisheries remaining in the food web after equilibrium is reached. |
| Overall effect indicator | Each indicator is rescaled between 0–1, and then the average of the five indicators is taken. |

levels (vertical distance) or trophic links (also accounting for horizontal distance). Therefore, we compare the following network scenarios: (3) Similar: harvesting similar trophic levels (Fig. 1B) resembling historical fisheries of high trophic level species[30] but also the growing number of fishmeal factories, for example, in West Africa[31], which target small-sized species; (4) Trophically-balanced: the target species chosen are evenly distributed across the trophic levels (Fig. 1B) resembling how humans place value on having a variety of fish species to eat[11,32,33], such as sprat (Sprattus), herring (Clupea), and cod (Gadus) (small, medium, large species, respectively), which represents a network-based analogy to balanced harvesting strategies[20,34,35]; and (5) Distanced: harvesting species with largest network distance between them (Fig. 1B) which is a theoretically-informed strategy to spread disturbances from resource extraction across the food web to minimize their additive effect[23] and reduce the likelihood of amplified trophic cascades[36,37]. In this research field, these network scenarios serve as a starting point for systematic characterization of potential exploitation scenarios.

We independently simulated these fishery scenarios in 800 complex food webs comprising 130 species each (see Methods for details). We assessed indicators of the ecological system's perturbation stability (species persistence and food-web biomass change, see Table 1) and economic viability (sustained total revenue, sustained total biomass catch, and the number of sustained fisheries, see Table 1). Species persistence and food-web biomass change are calculated relative to the pristine community without fisheries. To assess the overall ecological and economic effect, we additionally calculated an indicator combining all of them ("overall effect indicator", see Table 1). We expected that the fishery scenarios should lead to divergent consequences for ecological and economic indicators. For example, fisheries targeting larger species such as salmon, tuna, or cod, which usually fetch higher market prices per kilogram than smaller species despite similar fishing costs, may appear to be most economically profitable[38–40] but their harvesting could also affect ecological stability by strong trophic cascades[28,32,41], which should translate into reduced fishery yield overall. So far, however, the repercussions of fishing activities that are incentivized by short-term profitability on ecosystem dynamics have not yet been formally tested for multiple fisheries in complex food webs. It thus remains an open question if complex food web dynamics lead to different optimal short-term and sustainable long-term economic strategies.

## Results
### Performance of fishery scenarios
From an ecological perspective, the six fishery scenarios performed similarly in terms of species persistence (Fig. 2A, except for the slightly lower persistence under Prices) and community biomass (Fig. 2B). Instead, substantial variation in these response variables prevailed regardless of the scenario. Interestingly, part of this variation can be attributed to differences in biomass losses across trophic levels (Fig. 3). The fishery scenarios Prices and Distance primarily lead to biomass reductions in upper trophic level species. This aligns with

their respective targeting strategies: Prices focuses on high-trophic-level species that yield higher market prices, while Distance aims to harvest species across both high and low trophic levels to maximize the number of links between target species. In contrast, the other scenarios predominantly impact the biomass of mid-trophic-level species (Fig. 3). In consequence, the Similar scenario also leads to the most even distribution of the biomass catches across the fishery fleets (Supplementary Fig. S1). Despite generally very consistent economic performance of the six scenarios (Fig. 2C–E), we found a subset of the simulations of the Similar scenario, where species of similar trophic level are harvested, that performed better than the other scenarios. This subset of Similar supplied the largest fish biomass catches (Fig. 2C), yielded the highest mean total sustained revenue (Fig. 2D), and resulted in the highest average number of viable fisheries (Fig. 2E). Therefore, Similar also yielded the highest chances of maintaining all three fisheries (Fig. 4A, inlay plots, Random: 16%, Productivity: 0%, Prices: 0%, Similar: 49%, Trophically-balanced: 1%, Distanced: 0%). Remarkably, the two economically-inspired fishery scenarios manifest themselves as intricately defined subcategories within the broader framework of Similar, wherein Prices focuses on harvesting fish resources of high trophic levels (Fig. 4Aiii), while Productivity exhibits a consistent inclination towards exploiting low trophic levels (Fig. 4Aii). Overall, these results suggest that the Similar scenario can maximize economic yield and ecological resilience under specific conditions, while substantial within-scenario variation in the ecological and economic response variables remained unexplained.

### Viable fisheries and target species' trophic levels across fishery scenarios
Much of this variability arises from across-simulation differences in species traits such as body mass, trophic level, and productivity. Among these, body mass, apart from environmental factors such as nutrient availability and temperature, is the strongest driver of population productivity[42]. Body mass also establishes a feeding hierarchy within food webs, with larger species predominantly preying on smaller ones[43] and thus also determines trophic levels. As a result, the species traits of body mass, trophic level, and productivity are highly correlated in our models, reflecting similar patterns observed in natural ecosystems. Based on our ecological expectations, we assessed the variation in the ecological and economic response variables in terms of the mean and standard deviation of the harvested species' trophic levels (Figs. 4 and 5). The mean trophic level captures from which trophic level the three target species are harvested on average (i.e., the species harvested are characterized on a gradient from one for basal species to levels of often five or higher for top species).

The standard deviation of trophic levels reflects the similarity in trophic levels harvested with a low standard deviation indicating the fishing of similar trophic levels (as in the Similar scenario), and intermediate (Trophically-balanced scenario) and a high standard deviation (Distanced scenario) indicating increasing differences in the spread of the target species across trophic levels (Fig. 4A iv–vi). While the three network scenarios differ substantially from one another, the Prices and

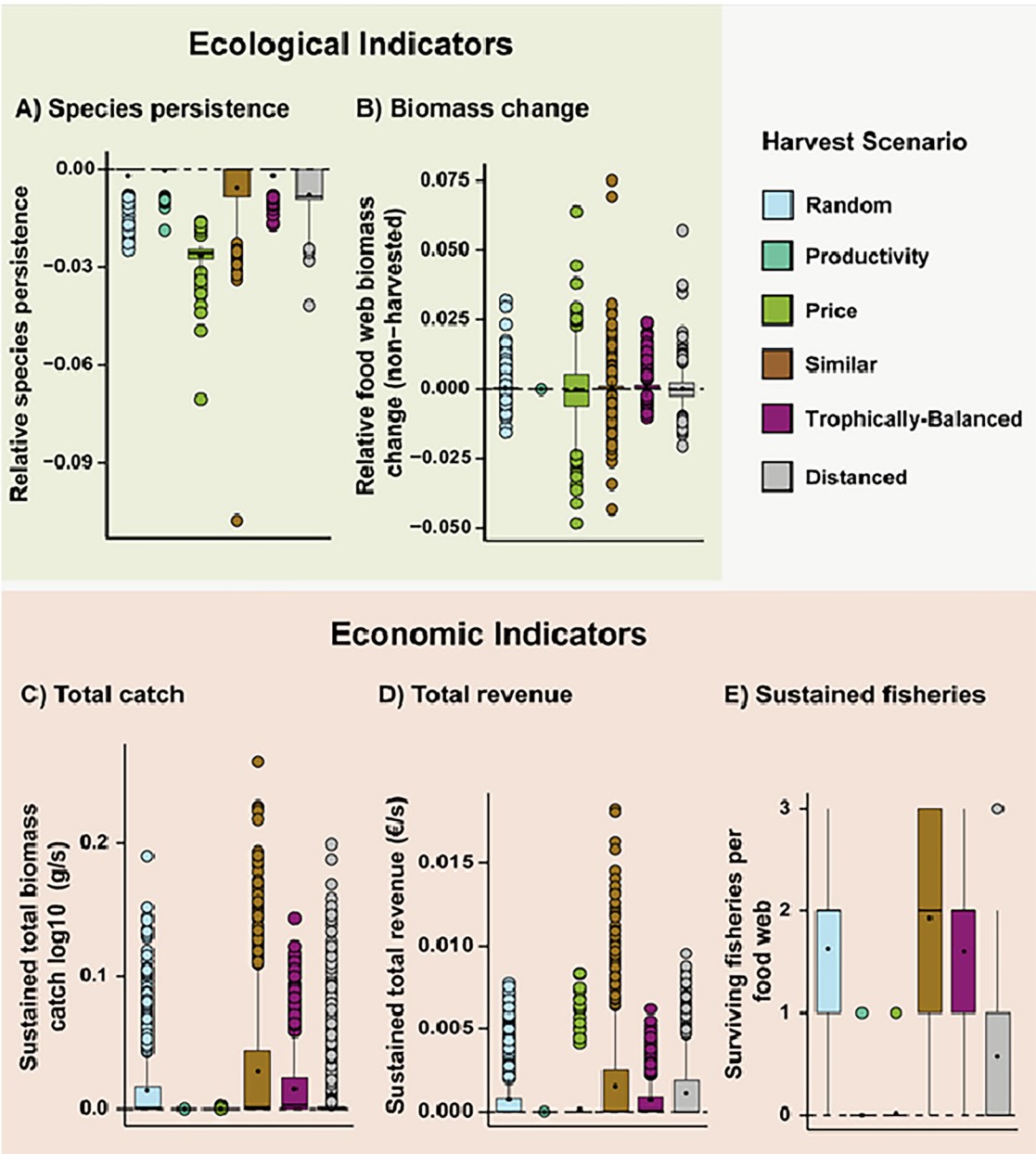

**Fig. 2 | The impact of fishery scenarios.** Five response variables at the food web level showing the impact of fishery scenarios (x-axis, colour code) (see Table 1 for definitions): **A** Species persistence; (**B**) Food-web biomass change, excluding harvested species; (**C**) Sustained total biomass catch by all three fisheries; (**D**) Sustained total revenue by all three fisheries and; (**E**) Sustained number of fisheries. The dot is the mean; the middle line of the boxplots represents the median and box limits correspond to the 0.25 and 0.75 quantiles. Whiskers represent the 1.5 inter-quartile range. The food webs containing the top and bottom 0.05 quantile were identified and removed for each indicator to aid in visualisation, leaving $n = 731$ food webs for each harvest scenario.

Productivity scenarios overlap with parts of the Similar scenario, specifically in targeting high (Similar high) and low trophic levels (Similar low), respectively (Fig. 4A ii and iii versus iv). However, the portion of the Similar intermediate scenario that focuses on intermediate trophic levels is unique and not shared with any other scenario (Fig. 4A iv).

With regard to the number of sustained fisheries, we found within the six scenarios (Fig. 4A) and across them (Fig. 4B) the general pattern that the highest probability of three sustainable fisheries only occurs when the trophic levels of the target fish species are similar (i.e., low standard deviation of target species trophic levels). This explains the advantages of the Similar scenario (Fig. 4Aiv) compared to the other scenarios (Fig. 4A) since it corresponds to harvesting species with a low standard deviation of their trophic levels, regardless of the mean. The Random scenario results in unconstrained variation in the mean and

standard deviation of fish trophic levels (Fig. 4Ai), leading to highly variable results. The fraction of runs where zero fisheries persisted varied widely between scenarios (Random: 10%, Productivity: 100%, Prices: 98%, Similar: 20%, Trophically-balanced: 4%, Distanced: 57%). This variability finds an explanation in the average trophic level harvested. Under the Similar scenario targeting fish species with similar trophic levels (i.e., a low standard deviation of trophic levels), we found a clear separation between fishing for mid-trophic level fish (Similar intermediate, trophic levels between 3 and 5), which leads to a high probability of three sustainable fisheries, and fishing for low or high trophic level fish (Similar low, high), which leads to a low number of sustained fisheries (Fig. 4Aiv). Since this pattern holds irrespective of the assumed fishery scenario (Fig. 4B vs 4Ai–vi), we did not separate fishery scenarios in the following analyses. Instead, we add a dashed

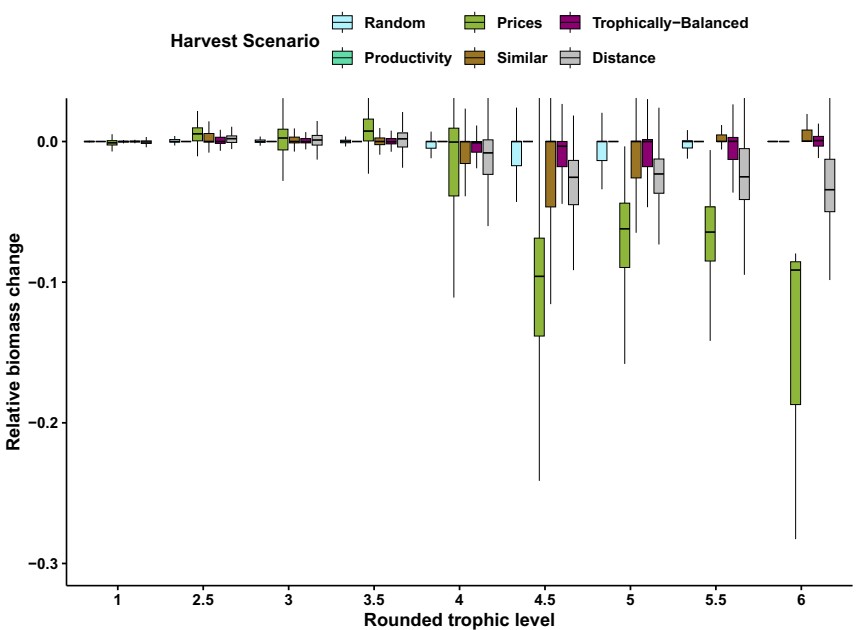

**Fig. 3 | Distribution of the biomass changes caused by the six fishery scenarios across the trophic levels of populations in the food webs.** Relative biomass changes represent the non-extinct population biomass densities at the end of the fisheries simulations compared to their initial densities before fisheries were introduced. The trophic levels of the populations are rounded to the nearest 0.5 for clarity. The middle line of the boxplots represents the median and box limits correspond to the 0.25 and 0.75 quantiles. Whiskers represent the 1.5 interquartile range. Number of data points for each scenario are 111,395 (Distance), 109,343 (Prices), 112,224 (Productivity), 111,934 (Random), 111,432 (Similar), 112,030 (Balanced).

line at a harvested trophic level standard deviation of 0.5 characterizing the maximum standard deviation of trophic levels of the Similar scenario (e.g., Figure 4B).

### Ecological resilience and economic profitability driven by target species' trophic levels

We found that harvesting species of similar trophic levels (i.e., low standard deviation) can cause both the fewest extinctions if low or intermediate trophic levels are targeted or the most extinctions if the highest trophic levels are harvested (Fig. 5A). Targeting the highest trophic levels also caused the highest instability (measured as the coefficient of variation of overall species biomass, (Supplementary Figs. S2, S3). The community biomass and also the total catch of the fisheries are highest if similar species of intermediate are harvested (Fig. 5B, C). These findings also hold when calculating the share of biomass caught compared to the biomass of all harvested species (Supplementary Fig. S4) or the total biomass of all species (Supplementary Fig. S5). The high species persistence and low biomass losses demonstrate that the ecological system's resilience is highest if fish at intermediate trophic levels are harvested (Fig. 5A, B). The higher sustained biomass translates into higher biomass catches of the fleets (Fig. 5C) and thus the highest total sustained revenue (Fig. 5D). Surprisingly, our results reveal no trade-off between ecological stability and sustainable economic revenue. Instead, targeting fish at intermediate trophic levels optimizes both of these tightly connected goals.

To integrate these similar, yet distinct, responses of the five ecological and economic indicators (Fig. 5A–D), we developed an overall effect indicator (i.e., the mean of the five indicators after scaling, see methods) showing the highest values are achieved when species of similar (low standard deviation) and intermediate mean trophic levels are fished (Fig. 5E, red color). The overall indicator also reveals that the system responds poorly when the fisheries target (1) similar (low standard deviation) low or high trophic level species or (2) species of very different trophic levels (high standard deviation). We carried out sensitivity analyses and found that these results are robust against

variation in whether the target animal is a vertebrate or an invertebrate (Supplementary Fig. S6), variation in the number of fisheries (Supplementary Fig. S7), removing our economic assumption that bigger fish are worth more per weight by using the same price for all fish species (Supplementary Fig. S8), and using the food web species richness prior to the pristine run instead of after the pristine run (Supplementary Fig. S9).

## Discussion

By integrating humans in the form of fisheries with the underlying dynamics of price and demand into a dynamic food-web model, we show that surprising win-win situations occur with high sustainable economic gain and low negative ecological impact arise when similar mid-trophic level species are caught in multispecies fisheries.

Our multi-fishery scenarios compare two wide-spread scenarios characteristic for unconstrained open-access fisheries that target the fish populations with the highest biomass productivity (Productivity) or the highest price on the market (Price) to scenarios constraining the target species according to their position in the food web. Harvesting multiple fish populations of similar trophic levels (Similar) represents fisheries that are constrained by a specific type of gear that only allows capturing species of specific body-size classes as the fisheries supplying the growing number of fishmeal factories, for example, in West Africa[31]. Targeting fish populations that are evenly distributed across the trophic levels (Trophically-balanced) is a simplified version of balanced harvesting strategies[20,34,35] and can also naturally arise if humans place value on having a variety of fish species to eat[11,32,33], such as sprat (Sprattus), herring (Clupea), and cod (Gadus) (small, medium, large species, respectively). A theoretically-informed strategy aims to separate the fishery disturbances as distant as possible to each other (Distanced) to minimize their additive effect on other populations. We hypothesized that harvesting species of similar trophic levels (Similar scenario) would result in the most negative ecological and economic impacts due to the additivity of disturbance effects and the erosion of energy flows through these trophic levels. Surprisingly, our analyses

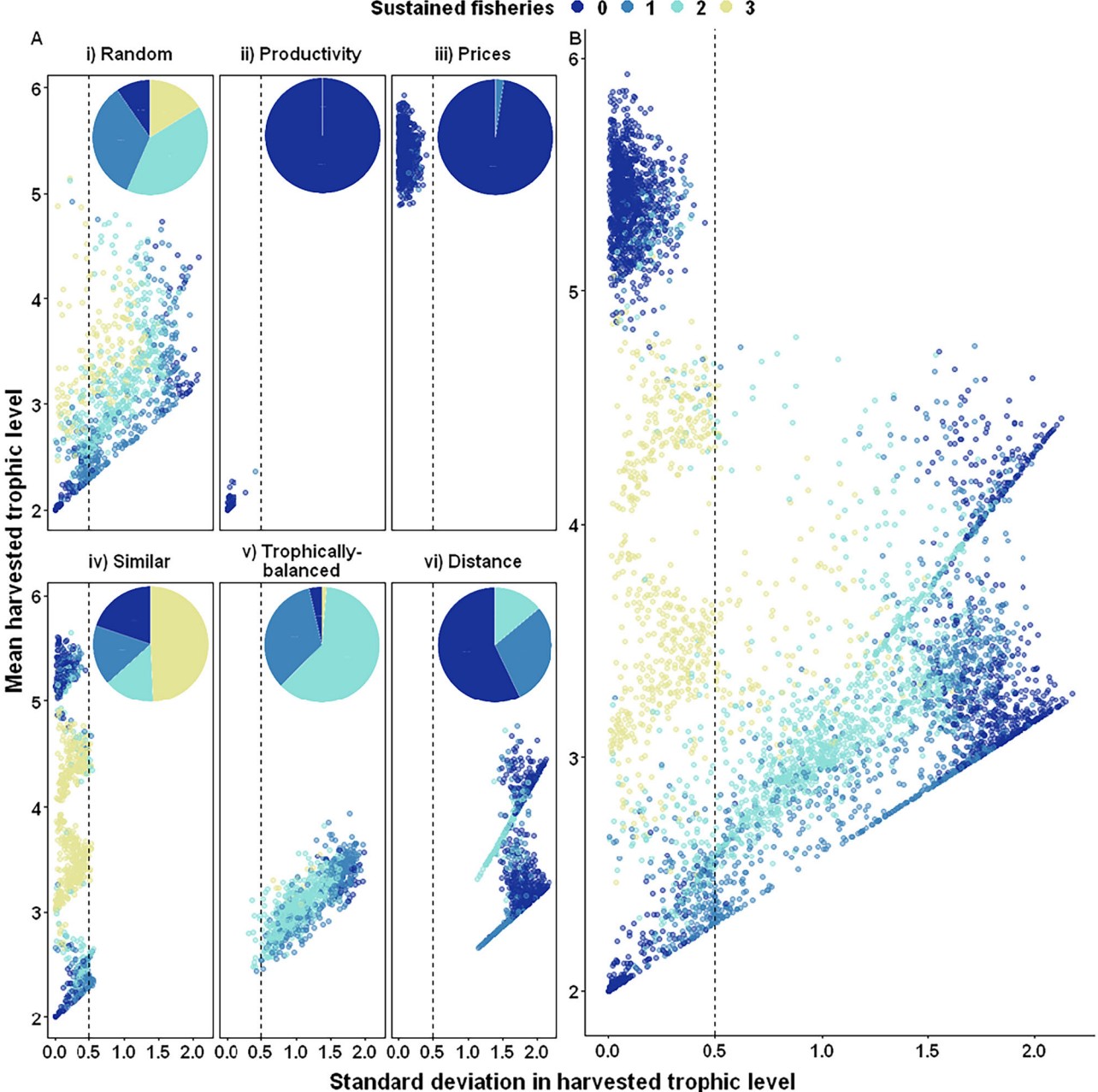

**Fig. 4 | Number of sustained fisheries (colour code) depending on the standard deviation in harvested species trophic level (x-axis) and the average harvested species trophic level (y-axis). A** Per scenario and **B** pooled for all scenarios. Each dot represents one food web and is characterised by the average and the standard deviation of the three target species' trophic levels. Inlay pie charts show the fraction of runs ending with zero to three sustained fisheries. The dashed line at x = 0.5 serves as a reference and represents the maximum standard deviation of the trophic levels of the Similar scenario.

did not support this hypothesis. Instead, this strategy leads to an overall positive effect on our indicators if fish of similar intermediate trophic levels are harvested. An explanation can be found in the effects a fishery has on the food-web neighborhood of its harvested species. Consistent with prior results[26], we found that the closest species to the harvested species (i.e., its prey and predators, respond the strongest with biomass increases and decreases, respectively (Supplementary Fig. S10A). Species of a similar trophic level likely have similar resources and consumers and are thus engaged in network motifs of exploitative and apparent competition, in which predator-mediated coexistence maintains biodiversity. Harvesting only one of these competitors creates a biomass imbalance, leading to increased competitive pressure, subsequent further losses in biomass, and eventually the extinction of the harvested species[44,45] (Supplementary Fig. S10B).

As our food webs are biomass heavy at the base and top (Supplementary Fig. S11) and have the highest interaction strengths or energy influx at lower trophic levels (Supplementary Fig. S12), harvesting these levels would cause the greatest disturbance to the food web. This may explain the negative ecological effects of the Trophically-balanced (fishing species are evenly distributed across trophic levels) and Distanced scenarios (species with the largest link distance to each other are harvested). In contrast, this imbalance in perturbation is less likely in the Similar scenario as the odds of harvesting both competitors are higher (Supplementary Fig. S10C), making it superior compared to the Trophically-balanced and Distanced strategies. The economically-driven scenarios represent subsets within the overarching Similar scenario, encompassing either the highest trophic level fish that command premium prices in the market under Prices[21], or the lowest

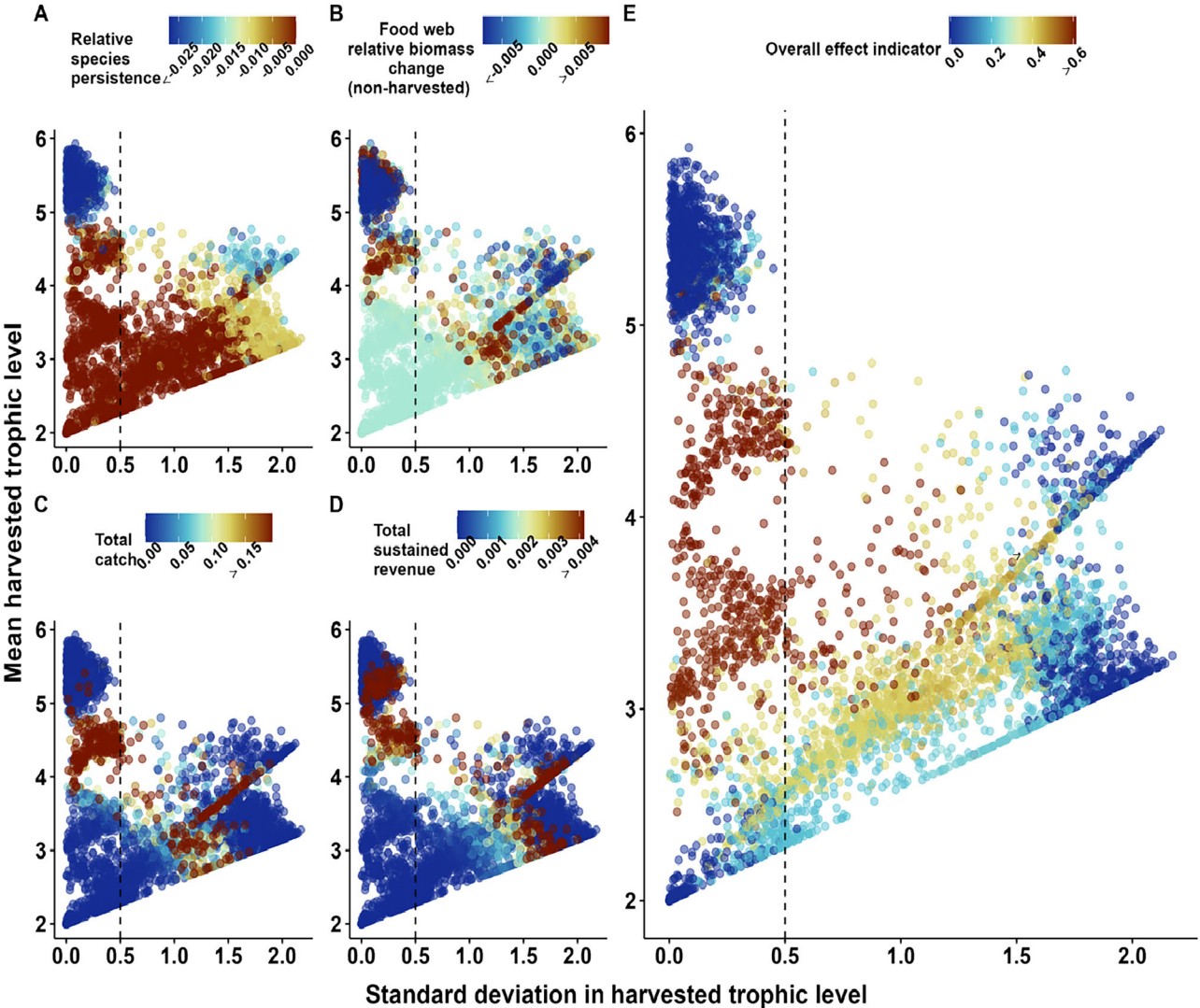

**Fig. 5 | Different indicators plotted by the standard deviation in harvested trophic level (x-axis) against the mean harvested trophic level (y-axis).** A Species persistence, defined as the ratio of surviving vs initial species count in a food web at the start of harvest; **B** Food-web biomass change, excluding harvested species, defined as the summed harvested biomass divided by the summed pristine biomass for non-harvested species in a food web; (**C**) Sustained total revenue, defined as the sustained revenue sum of all three fisheries; (**D**) Sustained total biomass catch, defined as the sum of biomass caught by all three fisheries and; (**E**) Overall effect indicator, defined as the mean of all self-scaled indicators. Each dot represents one food web. The dashed line at x = 0.5 is given for reference. The systematic linear patterns visible are a result of whether the middle species harvested is closer in trophic level to the smallest species (lower edge) or closer to the largest species (upper edge) (See Supplementary Fig. S15 for a visualisation of this effect).

trophic level species that exhibit the highest productivity under Productivity[20]. Collectively, these findings underscore the consistent importance of the target species' trophic level when evaluating the multifarious effects of multiple fisheries.

Consistent with prior empirical[27,28,41] and model studies[24,25], we found that harvesting species of high trophic levels is ecologically detrimental in terms of species extinctions, community biomass losses, and stability. Several factors explain the detrimental consequences of harvesting high trophic level species, including their higher sensitivity to perturbations[46–49], their potential to unleash trophic cascades with potential food web break down[50], and the lower number of species per functional group leading to a higher likelihood of functional loss[51]. Supporting our expectations, we also found that harvesting species from lower trophic levels led to less sustained fisheries[52], which can be explained by cutting off energy channels from the basal species to the rest of the community[29]. These results stem from the energy fluxes and biomass structure of our food webs. Biomass is produced at the base of the food web and transferred as energy through the links to

upper trophic levels. While stronger interaction strengths, implying higher energy fluxes, could reduce the vulnerability of high-trophic-level species to disruptions in energy supply, weaker interaction strengths would diminish the strength of trophic cascades, thereby reducing the sensitivity of low-trophic-level species to changes at upper trophic levels. This interplay suggests that our results could be influenced by the food-web environment. To account for this, we used food-web structures and interaction strength distributions that adhere to constrained empirical patterns[43,53,54]. Additionally, to reflect the natural variation found across communities, we incorporated random variation in food-web and interaction-strength parameters, keeping them within empirically observed limits. Despite this variability in model parameters, which is also visible in our response variables, our results consistently highlight the ecological implications. Specifically, the critical role of low-trophic-level species in sustaining energy flows and the risk of triggering trophic cascades by harvesting high-trophic-level species emphasize the ecological suitability of targeting intermediate trophic levels for fishing.

Not only is harvesting fish species of similar intermediate trophic levels more likely to buffer disturbances, but we have found that this also results in relatively high sustained catches, revenues, and fisheries. An increase in sustained revenues with the harvested fish trophic level is understandable based on simple economic principles, as bigger fish fetch a higher price per weight due to their natural scarcity. Despite higher catch costs, this explains the increase in revenue up to inter-mediate trophic levels. Above a certain trophic level, however, this positive price effect is slowed by a decline in community biomass as the trophic level of the fish harvested increases, resulting in a decline in the biomass of fish caught and, thus, a decline in fishery revenues and consequently a decrease in the number of viable fisheries. This indi-cates that the economic profitability of fishing in an open-access set-ting is a double-edged sword. Whereas the high price of larger fish can drive short-term fishery strategies, in the long term, profitability decreases due to decreasing fish abundance[55]. Here, we find that the best balance between these effects is obtained at medium trophic levels. These results show that the sustainability of economic resource exploitation strongly depends on the food web's ability to maintain biodiversity and biomass despite disturbance. Similarly, ecological resilience is determined by the economic dynamics of prices, fleet size, and, thus fishing intensity. Taken together, our results show a strong interaction between ecological and economic dynamics that produces an integrated attractor with maximum ecological resilience and eco-nomic benefits.

Our results are informative for fisheries managers because mul-tiple fisheries are present in most ecosystems. Therefore, our approach went beyond the more common single-species models[56,57] to show that the composition of targeted fish species matters for both ecological and economic outcomes. Furthermore, our results can also be informative for policymakers considering opening new fisheries in their systems. For instance, there is an intense debate on whether to start commercial harvesting of the small but highly abundant species in the mesopelagic zone[58,59], which could lead to the collapse of the traditional higher trophic level fisheries due to ecological feedback loops[58]. Our results suggest that harvesting lower trophic levels, especially of similar size, is potentially highly detrimental to ecosys-tems and fisheries. The most promising fishing strategy in our model analysis, the harvesting of similar populations of intermediate trophic levels, does not strictly follow the general balanced harvesting approach. Yet, the similarity with the balanced harvesting approach is that selective fisheries on the top of the food web should be avoided. Thus, potentially negative consequences of interactions among fish-eries depending on their network context constitute an urgent reason to coordinate management policies through an ecosystem-based approach that takes into account the complex web of interactions[60]. This is complicated by the fact that detailed knowledge of food web structures is available for only a few marine ecosystems[61]. However, our results related several ecological and economic indicator values to the mean and standard deviation of trophic levels of target species. The advantage of using trophic levels as indicators of the position of target species in the network is that they can be represented by stable isotope measurements of the target species as delta[15]. N relative to the baseline of the system[62,63]. This approach can also account for shifts in trophic levels within species during ontogeny by sorting individuals of a species according to stage-specific stable isotope signatures[64,65]. This widely available information on stable isotope signatures can therefore promote the broad application of our findings on managing multi-species fisheries using the information on fish trophic levels.

Our approach has realized a synthesis of economic multi-fishery models with dynamic models of complex food webs addressing a major criticism in fishery single-species management[66]. This yielded some results going beyond simple models of one or few species embedded in modules. Firstly, smaller food web modules cannot accurately represent natural communities and model impacts of

fishery on biodiversity and the community biomass distribution due to the low number of species. Secondly, our modelling approach of complex systems involves feedback loops that cascade through the network links. Ultimately, these feedback loops are one of the key drivers causing interactions between the multiple fisheries in our approach, leading to knock-on consequences for the sustainable yield and revenue. These advances have been achieved by several simplifi-cations that provide interesting opportunities for future research. First, we chose to model our fisheries as open-access though com-mercial catches come from managed and unmanaged (open-access) fisheries in the real world[66]. In open access, catch quantities are not controlled, and it would be interesting to study the consequences of active management (e.g., by maximum sustainable yield) that could prevent the harvesting of low-biomass species. In this vein, more complex approaches to modeling balanced harvesting, also taking fish productivity into account, could also provide important connections to real-world fisheries[20,34,35]. Second, we abstracted from economic interactions between species via coupled harvesting and consumer preferences[33]. For example, some species, such as Cod (Gadus) and Pollock (Pollachius), may be substitutable, or there may be social or cultural preferences for or against certain species. Third, we simplified our model by limiting each fishery to one species, and it would be interesting to implement more complex fishery strategies to harvest multiple species, if not simultaneously, then switching between spe-cies depending on the opportunity[67,68]. Fourth, our model could be extended to include the consequences of by-catch[69]. Our model pro-vides a framework for studying the effects of multiple fisheries in complex food webs that is flexible to incorporate these different components. We anticipate that much-needed future research may contribute to our understanding of how socioeconomic factors influ-ence complex food webs, leading to the development and incorpora-tion of ecosystem-based management to ensure that we can feed our growing population while maintaining ecological functioning.

Research on the impacts of fisheries and their management has evolved from single-species models[56,57] to multispecies models[24-26]. Here, we expanded this approach from single to multiple economically constrained fisheries. Our results highlight that sustainable economic gains cannot be understood without accounting for feedback via the stability of the ecological community that is perturbed by short-term economically more profitable strategies such as harvesting large top predators. Instead, we show that harvesting similarly-sized, mid-trophic-level species leads to some of the highest sustained food web biomasses and catch while supporting high food web persistence and sustained revenue. These surprising win-win situations with high sus-tained economic gain and low negative ecological impacts can only be derived based on complex systems analyses because theory on simple food chains or single-fishery models cannot account for the feedback effects through the networks. Our fishery exploitation scenarios represent a systematic, yet not exhaustive, set of multi-fishery strate-gies that can be adjusted to account for real-world variations in fish species, community complexity, and environmental gradients. Nevertheless, our results demonstrate that integrating ecological and economic dynamics into complex models is a necessary prerequisite for sustainably using natural resources to support the demands of a growing world population. The simplicity of our economic-ecological win-win situations can aid fishery managers in identifying combina-tions of established or new open-access fisheries that are ecologically and economically sustainable. Complex systems analyses can there-fore provide guidance on how to improve the sustainable supply of fish protein to the world's growing population to prevent an increase in world hunger without risking damage to fragile ecological systems.

## Methods
Our approach integrates a dynamic allometric food web model and an economic model of resource exploitation. The parameters of the food-

web models are chosen based on empirical scaling relationships (see Supplementary Tables 1–4).

## Food web topology

Each food web initially contains 30 primary producer species (plants) and 100 consumer species, with 67 invertebrates and the 33 remaining being fish. $log_{10}$ body masses, $m_i$, are generated from a uniform distribution of [0,6] for plant species[70], [2,6] for invertebrates and [4,10] for fish[70]. To generate realistic food webs, based on biologically informed rules, we used a method that builds on the empirically established importance of body mass relationship between consumers and resources, especially relevant in aquatic ecosystems[43]. The trophic niche of each species is defined as a body mass range of resources upon which a consumer can feed on and is determined with a Ricker function. The Ricker function quantifies the likelihood ($L_{ij}$) of consumer $i$ successfully capturing and eating resource $j$ based on their respective body masses $m_i$ and $m_j$.

$$L_{ij} = \left( \frac{m_i}{m_j R_{opt}} e^{1 - \frac{m_i}{m_j R_{opt}}} \right)^{\gamma} \tag{1}$$

$R_{opt}$ defines the optimal consumer-resource body mass ratio and $\gamma$ the width of consumer trophic niche. The Ricker function produces a hump-shape response curve meaning that consumption likelihood is the biggest in the vicinity of $R_{opt}$ and decreases when the body mass of the resource goes away from it. If $L_{ij}$ is less or equal to 0.01, it is set to zero to indicate no interaction[70]. The resulting food webs are non-random in terms of the ecological mechanisms that determine their links, while generality is achieved through a replicated random distribution of the community body masses.

## Ecological dynamics

Trophic relationships (consumer and resource) and metabolism drive consumer biomass dynamics.

The two limiting nutrients, $N_l$, for which plants compete and which they store within their biomass, $B_i$, are constantly refreshed. Nutrient dynamics are calculated as

$$\frac{dN_l}{dt} = D(S_l - N_l) - v_l \sum_i r_i G_i B_i \tag{2}$$

Where the nutrient replenishment rate is $D$, and $S_i$ is the maximum nutrient concentration. Nutrients are lost as they are taken up by plants, which depends on the relative nutrient content in the plant's biomass $v_l$, and plant growth, scaled by their intrinsic growth rate $r_i$, where $r_i = m_i^{-0.25}$ and specific growth factor $G_l$.

$$G_i = \min\left( \frac{N_1}{K_{i1} + N_1}, \frac{N_2}{K_{i2} + N_2} \right) \tag{3}$$

We calculate $G_i$ using the nutrient concentration $N_i$ and the nutrient uptake efficiency, which is determined by the half-saturation density $K_{il}$[70],.

The plant and animal dynamics are defined as:

$$\frac{dB_i}{dt} = r_i G_i B_i - \sum_k B_k F_{ki} - x_i B_i \tag{4}$$

for plants
and

$$\frac{dB_i}{dt} = B_i \sum_k e_k F_{ik} - \sum_k B_k F_{ki} - x_i B_i \tag{5}$$

for animals.

Where $e_k$ is the assimilation efficiency, different for plant and animal resource species[71] and metabolic demands $x_i$ is allometrically defined: $x_i = y_i m_i^{-a}$.

## Feeding rates

The feeding rate of consumer $i$ on resource $j$, $\mathbf{F_{ij}}$, we calculate as

$$F_{ij} = \frac{\omega_i b_{ij} B_j^{1+q}}{1 + c_i B_i + \omega_i h_i \sum_k b_{ik} B_k^{1+q}} * \frac{1}{m_i} \tag{6}$$

The Hill exponent, $1 + q$, is specific for each food web[70], $\mathbf{c_i}$ defines the consumer interference. The handling time, $\mathbf{h_i}$, depends on consumer and resource body sizes:

$$h_{ij} = h0 * (m_i^{\eta_i} * m_j^{\eta_j}) \tag{7}$$

The relative consumption rate $\mathbf{\omega_i}$, is defined as one over the number of resource species of consumer $i$. The resource-specific capture coefficient $\mathbf{b_{ij}}$, equals the product of movement-dependent encounter rates and $\mathbf{L_{ij}}$ (equ. 1):

$$b_{ij} = b0_{herbivore} m_i^{\beta_{consumer}} L_{ij} \tag{8}$$

herbivore

$$b_{ij} = b0_{carnivore} m_i^{\beta_{consumer}} m_j^{\beta_{resource}} L_{ij} \tag{9}$$

carnivore

where $\beta_{consumer}$ and $\beta_{resource}$ are allometric scaling exponents, and $b0_{herbivore}$ and $b0_{carnivore}$ represent the intercepts of these scaling relationships. These expressions differ for herbivores and carnivores because herbivore-plant encounters only depend on herbivore movement, whereas encounters amongst animals depend on consumer and resource movement.

## Human fisheries as food-web nodes

In our main analyses, fisheries are targeting three fish resources that are chosen according to six scenarios (Fig. 1) to maximize (1) their price (Price scenario): the three species with the highest prices are selected, (2) their productivity (Productivity scenario): the three species with the highest productivity are selected, (3) their distance in terms of the number of network links between them (Distanced scenario): the combination of 3 species maximizing the sum of links between them is selected, which also minimizes (i) the likelihood that the species are connected and (ii) the redundancy of their roles in the food web, (4) their distance in terms of trophic levels (Trophically-balanced scenario): the combination of three species maximizing the difference in trophic levels between them is selected, which are compared to harvesting (5) the most similar target species in terms of their trophic levels (Similar scenario: Similar high: with trophic levels greater than 5, Similar low with trophic levels lower than 3, or Similar intermediate with grouped trophic levels between 4 and 5 or between 3 and 4), and (6) randomly targeted species ('random' scenario). Species productivity was defined as the difference between energy influx and metabolic rate, which corresponds to the amount of energy available for biomass increase in absence of predation. This calculation has been made at the end of the pristine simulations and therefore represents a fixed property of the ecosystem before the fishery disturbances. The choice of the three target species is clearly defined in all cases, except for the choice of the lowest trophic level target species, where multiple species could be herbivorous and have the same trophic level of two. In these cases, we randomly selected the target species amongst herbivores.

These fisheries affect the food web by harvesting a total amount of biomass stemming from a targeted species $i$, $\mathbf{catch_i}$, through vessels in

a fleet, $V_i$. We assume an open access system with each fishery fleet consisting of identical vessels searching for and harvesting a specific fish species. Vessels enter or exit a fishery depending on whether fishing is profitable or incurs losses[72,73]. The initial vessel density, $V_i$, is randomly drawn from [0.0001,0.001] per fishery[25].

For the harvested species, the biomass dynamics (derived from equ. 5) are given by

$$\frac{dB_i}{dt} = B_i \sum_k e_k F_{ik} - \sum_k B_k F_{ki} - x_i B_i - catch_i \qquad (10)$$

where

$$catch_i = catch_{max i} \frac{{}^*B_i}{\frac{catch_{max i}}{clearance_i} + B_i} * V_i \qquad (11)$$

Catches increase with vessel density, $V_i$, the vessel maximum catch rate, **catch_max** $_i$, the surface area a vessel covers when searching for the captured species, **clearance** $_i$ (see SI 13 and Supplementary Fig. S13 for a derivation of the starting values), and with the biomass density of the target species, $B_i$. The actual catch per unit of vessel (i.e., $V_i$), is a concave function of the target fish species biomass, which captures a moderate degree of hyperstability, i.e. with declining biomass, catches decrease less than proportionally, consistent with empirical evidence[74].

The vessel density $V_i$ changes depending on profitability, i.e. the difference between revenues of selling catch on the market and fishery-associated costs (derivations follow in equ. 12-18). The per vessel cost of the fishery harvesting species $i$ (**cost$_i$**) consists of a maintenance cost (**maintenance$_i$**) that is independent of the vessel activity, an activity cost that depends on the amount of area covered while fishing (**clearance$_i$**), and of the equipment used on board (*gear*). We finally used a variable *scaling* to control the relative importance of the fishing area covered and maintenance costs on the per vessel costs:

$$cost_i = maintenance_i + scaling * clearance_i * \frac{1}{gear} \qquad (12)$$

Revenues are the product of catch (**catch$_i$**) and the market price of species $i$ (**p$_i$**). The market equilibrium price is determined by the market equilibrium condition that supply **catch$_i$** equals demand, which we assume to be a downward-sloping function determined by

$$p_i = \left(\frac{catch_i}{p_{base i}}\right)^{\frac{1}{PED}} \qquad (13)$$

with scaling parameter **p$_{base\ i}$** (equ. 14) and the price elasticity of demand, $PED = -1.15$, which shows how sensitive the species' market price is to a change in quantity[75]. As larger fish tend to be valued higher by the market[38,40,76], we adjust **p$_{base}$** to species $i$ by multiplying it with the ratio of $m_i$ over the mean consumer body mass $m_n$ in the food web (Supplementary Fig. 14).

$$p_{base i} = p_{base} * \left(\frac{m_i}{\bar{m}}\right) \qquad (14)$$

Fleet revenue, **revenue$_i$**, thus is

$$revenue_i = p_i * catch_i \qquad (15)$$

And the entire fishing fleet's profit, **π$_i$**, is

$$\pi_i = revenue_i - (cost_i * V_i) \qquad (16)$$

The change in vessel density and the clearance rate occur dynamically based on profit, increasing when profit is positive and decreasing when it is negative. Profit equals to zero once the steady-state or equilibrium is achieved.

$$\frac{dV_i}{dt} = \mu^* \pi_i \qquad (17)$$

for vessel density

$$\frac{dclearance_i}{dt} = \mu_c^* \pi_i \qquad (18)$$

for clearance rate

How quickly $V_i$ and **clearance** $_i$ respond to a change in $\pi_i$ is based on $\mu_v = 0.01$ and $\mu_c = \mu_v * 30$, respectively. We assume vessel density changes at a lower rate than **clearance$_i$** because **clearance$_i$** is associated with daily fishing activities, while the vessel density, $V_i$, can only change about once a month.

## Simulation set-up

We generated food webs with 100 initial animal species, ensuring every primary producer had at least one consumer and every consumer had at least one resource. These pristine food webs ran without harvesting for 100 years of model equivalence, allowing population stabilization to a steady state as a starting point for fishery simulations. Started densities for species were drawn from a uniform distribution between 0 and 10. A species became extinct when its biomass fell to very low densities, here chosen to be $10^{-6}$. Food webs were discarded if less than 75% of fish species remained at the end of this first simulation. The final pristine food webs composed of persistent species (thus excluding species that became extinct during the pristine simulations) and their biomasses were used as a starting point for the harvesting simulations. Final species biomass values from these pristine food webs are used to benchmark against the final species biomass values from the following harvested simulations. During harvesting, a fishery was deemed unsustainable and left the food web if its density fell to $10^{-10}$. The harvesting simulation ran for 100 years of model equivalence. Each food web simulation (food web generation, running the pristine and harvest scenarios) was limited to a maximum runtime of 24 h. A total of 800 food webs were generated, with each food web undergoing six harvesting scenarios (i.e., Productivity, Prices, Similar, Distanced, Trophically-balanced, Random, Fig. 1), resulting in 4800 data sets. Only the final results of the harvested scenario after reaching a steady state are compared to the final results of the pristine scenario, and neither revenue nor total catch is discounted (future values are weighed equally to present ones).

## Data availability

Data sharing is not applicable to this article as no empirical datasets were generated or analysed.

## Code availability

Code is available under  https://doi.org/10.5281/zenodo.15318757 (Ref. 77).

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

## Acknowledgements
This study was supported by the German Centre for Integrative Biodiversity Research (iDiv) Halle-Jena-Leipzig funded by the German Research Foundation by grant no. FZT 118 (A.S.W., M.R.H., R.R., K.L., G.A., M.Q., B.G., U.B.) and BR 2315/25-1 (U.B.), as well as the MultiTroph Research Unit and International Research Training Group TreeDì, jointly funded by the DFG and the University of Chinese Academy of Sciences (UCAS) by grant no.GRK 2324 and FOR 5281 (G.A.).

## Author contributions
Conceptualization: A.S.W., M.Q., B.G., U.B. Methodology: A.S.W., G.A., K.L., M.Q., C.R., B.G., U.B.; Investigation: A.S.W., G.A., B.G., U.B.; Visualization: A.S.W., M.R.H., R.R., B.G., U.B. Supervision: M.Q., U.B., B.G.; Writing—original draft: A.S.W., B.G., U.B.; Writing—review & editing: A.S.W., M.R.H., R.R., K.L., G.A., M.Q., B.G., U.B.

## Funding

## Competing interests
The authors declare no competing interests.
