## [Transparent Peer review file · Nature Communications]

Maintaining ecological stability for sustainable economic yields of multispecies fisheries in complex food webs

Corresponding Author: Professor Ulrich Brose

Version 0:

Reviewer comments:

Reviewer #1

(Remarks to the Author)
Comments to the author:

The authors present an interesting theoretical exploration of the role that different fisheries scenarios play in the economic viability of three fisheries and the stability of the harvested food webs. The manuscript is well written and focuses on the increasingly important issue of moving away from single species fisheries management towards broader multi-species perspectives that better capture the complex nature of these harvested food webs. The fishing scenarios used seem to be logical starting points, although the trophically balanced scenario may not quite capture the essence of what is commonly thought of as balanced harvesting management practice (i.e., across trophic levels proportionally with respect to their productivity). That being said, the food web analysis is logical and sound as a 'tip of the iceberg' exploration and the authors discuss many likely fruitful avenues for future work. I would suggest that the authors emphasize more throughout that this is a first foray into novel territory exploring the economic and ecological ramifications of harvesting in a complex multispecies food web framework. While the results are intriguing it is likely that it may be too early to make concrete conclusions as to what harvesting methods provide the greatest balance between economic profit and food web sustainability based on these initial results. My few comments below largely focus on the food web harvest scenarios as well as the figures which could be improved to ensure clear scientific communication of the concepts and results. I have not commented on the economic modelling/aspects of the manuscript as this application is a bit outside of my field. Overall, I think that this is an important manuscript that deserves to be published for a wide audience as it is an important stepping stone towards understanding multispecies management in harvested ecosystems.

Major and Specific Comments:

Ln 78-79 – The authors bring up the concept of trophically-balanced harvesting here as a practice that is simply evenly distributed across trophic levels. While this is a logical way to go about modelling this, it may confuse readers with the idea of balanced harvesting practices which go further than this (Garcia et al., 2012; Jacobsen et al., 2014; Zhou et al., 2010). Please make this clear to the reader and consider adding this concept into the discussion as another important avenue for future work.

Ln 80-81 – Here, the authors are referring to how the distance between species nodes in a food web may contribute to reducing the impact of species deletions. While this is logical and based on a solid theoretical foundation, the reader is wondering if there are any empirical examples of this theory being applied through management actions. It may be worthwhile to include refs here if so and if not emphasize that this is based solely on theory and has not been empirically implemented.

Figure 1 – The reader found panel C in this figure to be confusing. The trophic level spread could almost be misinterpreted as the horizontal axis of trophic spread (i.e., habitat coupling) instead of the intended vertical variance around the mean trophic level. It may be helpful for the reader if the authors were to change their language here or to link panel C to panel B more explicitly by having boxes cover the mean/variance within panel B. The authors should also change the title of B (ii) to trophically balanced to maintain consistency with how this is being modelled and avoid confusion with balanced harvesting practices (see comment above Ln 79-80).

Figure 2 – These bar charts are a little hard to look at as they are quite small. The authors may consider stacking the ecological indicators above the economic indicators and moving the legend to the top so that the reader is able to more clearly see the results.

Figure 3 & 4 – Here in panels A, the authors have shown how relative species persistence changes with mean and sd of harvesting. As the pristine runs have up to 25% difference in the number of species and the dynamics of these webs may thus be different, it may be worthwhile to include the raw species persistence in the supplementary material.

Ln 329 – “the” is repeated

References

Garcia, S. M., Kolding, J., Rice, J., Rochet, M.-J., Zhou, S., Arimoto, T., Beyer, J. E., Borges, L., Bundy, A., Dunn, D., Fulton, E. A., Hall, M., Heino, M., Law, R., Makino, M., Rijnsdorp, A. D., Simard, F., & Smith, A. D. M. (2012). Reconsidering the Consequences of Selective Fisheries. *Science*, 335(6072), 1045–1047. <https://doi.org/10.1126/science.1214594>

Jacobsen, N. S., Gislason, H., & Andersen, K. H. (2014). The consequences of balanced harvesting of fish communities. *Proceedings of the Royal Society B: Biological Sciences*, 281(1775), 20132701. <https://doi.org/10.1098/rspb.2013.2701>

Zhou, S., Smith, A. D. M., Punt, A. E., Richardson, A. J., Gibbs, M., Fulton, E. A., Pascoe, S., Bulman, C., Bayliss, P., & Sainsbury, K. (2010). Ecosystem-based fisheries management requires a change to the selective fishing philosophy. *Proceedings of the National Academy of Sciences*, 107(21), 9485–9489. <https://doi.org/10.1073/pnas.0912771107>

(Remarks on code availability)

Reviewer #2

(Remarks to the Author)

Review of “Maintaining ecological stability for the sustainable economic yield 2 of multispecies fisheries in complex food webs”

The authors use a network approach to understanding the system-wide impacts of multi-species fisheries under a variety of fishing scenarios (different target species) and find that due to complex interactions between ecological and economic dynamics, and the cascading effects of harvesting throughout food webs, selectively targeting similar intermediate-trophic-level species can provide a win-win scenario that benefits fisheries sustainability and revenue along with ecological resilience. This is an important result, and I think more notably, the whole food web approach of this study is an important contribution to the field. The authors’ approach specifically addresses scenarios where multiple independently-operated fisheries (i.e., each with their own market dynamics) harvest from the same food web. This is an interesting and novel approach that nicely falls in between classical single species fisheries approaches and those that consider harvesting impacts of size- or trophic guild-focused fisheries that often ignore individual species.

While I think this type of approach is needed in the literature, is timely, and the results here deserve to be communicated to a broad audience, I do feel as though I was left wanting a little more information on the intersection between the fisheries and the ecological impacts, and I found that the results may be somewhat hard to decipher for a general audience. I do however think that this could be addressed by the authors relatively easily and therefore the manuscript certainly has potential to be a high-impact paper. I will elaborate more on these thoughts below.

General comments:

In general, while the results were nicely described in the text, I felt that I was left wanting a little bit more information on the ecological effects of some of the results. One way to address this may be to create a more succinct summary figure of the key results (even if just for the Similar scenario focused on so heavily here) with clear signposting highlighting what is going on. One initial thought I had related to this, and perhaps an example of how to summarize part of the results, is that I appreciated SI 9 showing the biomass structure of the unexploited community, but why not contrast this with the change in biomass structure under a few key scenarios? In my opinion this alone would go a long way in communicating the ecological effects of the fisheries at scale of the whole food web. Another suggestion would be to create a summary figure better highlighting the mean trends in key situations (i.e., the Similar scenarios) to describe important results such as that discussed in Lines 197-203.

In a similar sentiment, I found myself wanting to understand some of the results more deeply and did not feel that I was provided the information to do so. In terms of the food web effects of harvesting, for example, I am curious how sensitive the sustainability of intermediate trophic level fisheries – and the resulting food web resilience – is to the strength of interactions (and therefore structure) in the underlying food web? Presumably, weaker average interaction strengths and a less top-heavy food web would be more susceptible to fishing selectively at both top and intermediate trophic levels (since there is less energy available to top predators). But is this a matter of the general results holding consistently, or would a switch to lower trophic level fisheries be deemed more sustainable under this scenario? Sensitivity/robustness to factors such as this were not explored very thoroughly, in my opinion, and yet it is these fundamental patterns that are needed for a

generalizable fisheries theory. A question remains: are these results robust in spite of the food web structure or because of it?

It would be useful to contrast your results with other proposed harvest strategies in the literature, along with real world scenarios. As an example (and certainly not the only option), “balanced” fisheries have been suggested as a way to preserve community structure. You also indirectly come close to this rationale (though through a different mechanism) when discussing the impacts of selecting for similar intermediate TP species and, in a way, balancing competitive interactions. But balanced fisheries have been met with a lot of criticism due to the unlikely feasibility of perfectly matching catches to production of all species in a food web (which is a dynamic property). Do any of your results speak to this or other strategies suggested/patterns seen in the literature (or the real world)?

Similarly, can you give examples of where different fisheries approaches may typically be seen, and if so can you make any inferences about their sustainability?

Next, I will outline some questions I was left asking myself, that perhaps the authors could address through further visualization, additional discussion, or supplemental information:

- How does the equilibrium catch or effort of different fishing scenarios look relative to the community structure? Being able to visualize, for example, different trait distributions (body size, trophic level, productivity, etc.) of the community, and where harvesting occurs relative to this, would be a helpful supplement to Figure 1 and allow for better interpretation of the community-wide harvesting impacts. This could also be extended to a summary of the economic traits of the targeted species.
- Similar to above, what is the evenness of the catch or effort in the different scenarios?
- Are there trade-offs between resilience and revenue in the Similar outcome, or does this scenario optimize both? (see approx Line 111)

Overall, I think the feeling of wanting to understand the drivers of the different outcomes, and therefore the robustness/generality of the conclusions drawn, are reflected in these comments and remaining questions. I do not think it would take a substantial amount of work to address these, but some notable effort into elaborating on these results (and communicating/visualizing them) would greatly improve this paper and explain the importance of these results more broadly.

Minor comments:

Table 1 and main text: try and make sure wording is consistent

Line 78: is the magnitude of catch (biomass) or the rate of exploitation distributed evenly?

Page 2 in general:

- Clarify in the text that the metrics are relative to an unfished community
- It may be useful to include data on relationships between different fishing strategies. For example, price-driven harvesting and the Similar scenario here with focus on top trophic levels are likely harvesting similar species.

Line 203: Increase in number of viable fisheries? Should this be decrease?

Line 327: Distanced scenario – does this unintentionally select for species that are not highly connected within the food web, or ones for which there is little redundancy for species that do interact with them? Could this be a driver of the lack of fisheries sustainability in this scenario? Similar to my comment above about trait distributions, having a better idea of how harvesting is distributed throughout food webs may be very insightful.

Line 330 and potentially within figures if possible: when referring to the different target trophic groups within the Similar scenario, referred to here as (5), perhaps name these 5a, 5b, etc.

Missing details in Methods (sorry if I missed any of these!):

- Rationale for values or distribution of important parameters such as γ and R_{opt} may be helpful.
- Some parameters are not defined in the Methods. Examples include $b0_{herbivore}$ and $b0_{carnivore}$, β
- Please explain how you are defining productivity and if it can change with fishing pressure and changes in food web structure. If your measure is indeed dynamic (and potentially perpetually changing if dynamics are oscillatory), how do you account for this?
- Some explanation and justification on how all of the values for costs, clearance, P_{base} , etc. were chosen would be helpful, as would knowing if these affect the robustness of the results in any way.
- Line 390: “capped to run up to 24 hours” – do you mean of computation time? This was not immediately clear to me

Figure 1: consider numbering/labeling the choice components in Fig 1A (random, productivity, price) as such and have this match the order presented in the caption.

- Additionally, here and in the text where relevant, consider referring to the “Balanced” scenario as “trophically balanced” or some variation to differentiate between this and what is already known as balanced harvesting in the literature.

(Remarks on code availability)

Version 1:

Reviewer comments:

Reviewer #1

(Remarks to the Author)

Reviewer #1 (Remarks to the Author):

The authors have done a thorough job incorporating reviewer feedback for an improved manuscript and I have no further major comments. I appreciate the authors efforts to link the theoretical expectations of species removals to empirical work and agree that the lack of evidence is likely due to logistical challenges. I have one final thought, that does not need to be incorporated into the manuscript but may be interesting as an avenue for further research. I am curious as to how adding an adaptive foraging response may act to alter these results. For example, in the face of species losses due to harvesting, species may be able to respond by prey switching that can have important consequences for food web stability (Valdovinos et al. 2010). I also have one minor comment for Ln 256, consider changing two-sided sword to double-edged sword. Overall, I believe that the manuscript is a well-written investigation into complex harvested food webs and is a timely and novel contribution that will be of interest to a broad audience.

Valdovinos, F.S., Ramos-Jiliberto, R., Garay-Narváez, L., Urbani, P. & Dunne, J.A. (2010). Consequences of adaptive behaviour for the structure and dynamics of food webs. *Ecology Letters*, 13, 1546–1559.

(Remarks on code availability)

Reviewer #2

(Remarks to the Author)

The authors seemed to address each of my previous comments, as well as the other reviewer's, sufficiently and I appreciate the consideration and time put in to reflecting upon them critically. This revision was an enjoyable read - well done.

(Remarks on code availability)

Reviewer #1 (Remarks to the Author):

The authors present an interesting theoretical exploration of the role that different fisheries scenarios play in the economic viability of three fisheries and the stability of the harvested food webs. The manuscript is well written and focuses on the increasingly important issue of moving away from single species fisheries management towards broader multi-species perspectives that better capture the complex nature of these harvested food webs. The fishing scenarios used seem to be logical starting points, although the trophically balanced scenario may not quite capture the essence of what is commonly thought of as balanced harvesting management practice (i.e., across trophic levels proportionally with respect to their productivity). That being said, the food web analysis is logical and sound as a 'tip of the iceberg' exploration and the authors discuss many likely fruitful avenues for future work. I would suggest that the authors emphasize more throughout that this is a first foray into novel territory exploring the economic and ecological ramifications of harvesting in a complex multispecies food web framework. While the results are intriguing it is likely that it may be too early to make concrete conclusions as to what harvesting methods provide the greatest balance between economic profit and food web sustainability based on these initial results. My few comments below largely focus on the food web harvest scenarios as well as the figures which could be improved to ensure clear scientific communication of the concepts and results. I have not commented on the economic modelling/aspects of the manuscript as this application is a bit outside of my field. Overall, I think that this is an important manuscript that deserves to be published for a wide audience as it is an important stepping stone towards understanding multispecies management in harvested ecosystems.

Answer: We greatly appreciate the constructive feedback on our manuscript and fully agree that this represents an early exploration of a novel field. In response, we have added cautionary statements to both the introduction and conclusion (lines 84-85, 317-319), emphasizing that the sustainable multi-fishery scenarios we tested should be regarded as a systematic starting point for further evaluations. These additions highlight that this research area is in its infancy and that adaptations to real-world conditions should be carefully considered.

Major and Specific Comments:

Ln 78-79 – The authors bring up the concept of trophically-balanced harvesting here as a practice that is simply evenly distributed across trophic levels. While this is a logical way to go about modelling this, it may confuse readers with the idea of balanced harvesting practices which go further than this (Garcia et al., 2012; Jacobsen et al., 2014; Zhou et al., 2010). Please make this clear to the reader and consider adding this concept into the discussion as another important avenue for future work.

Answer: This is an excellent point, which we have incorporated into the introduction of our fishery scenarios (lines 78-81) and the discussion of future directions (lines 294-295). We are grateful for this suggestion, as it has helped to strengthen the connection between our work and real-world fishery scenarios.

Ln 80-81 – Here, the authors are referring to how the distance between species nodes in a food web may contribute to reducing the impact of species deletions. While this is logical and

based on a solid theoretical foundation, the reader is wondering if there are any empirical examples of this theory being applied through management actions. It may be worthwhile to include refs here if so and if not emphasize that this is based solely on theory and has not been empirically implemented.

Answer: This is a very reasonable comment, but we acknowledge that it is also a challenging one to address. Evaluating whether the effects of ecological impacts decay with the number of links in a network requires manipulating populations and tracking their effects on many (if not all) other populations within a known food web. While this is feasible in simulations of model food webs (e.g., Berlow et al. 2009), it is logistically extremely demanding in natural systems.

As such, our hypothesis and the fishery strategy Distanced are primarily based on food-web theory. In response to this comment, we conducted a search for whole food-web studies investigating indirect effects but were unable to find a convincing empirical example. This lack of evidence likely reflects the significant logistical challenges associated with conducting such experiments.

To address this gap and strengthen the manuscript, we have added two references to meta-studies that summarize empirical findings on the strength of trophic cascades. These studies (Borer et al. 2005; Su et al. 2021) suggest that indirect effects typically extend over only one or two links from the source of a disturbance. We have updated the relevant section of the introduction on the fishery strategies (lines 81–84) to include these references, providing additional empirical support to complement the theoretical framework.

Berlow, E. L. et al. Simple prediction of interaction strengths in complex food webs. *Proc. Natl. Acad. Sci.* 106, 187–191 (2009).

Borer, E. T. et al. What Determines the Strength of a Trophic Cascade? *Ecology* 86, 528–537 (2005).

Su, H. et al. Determinants of trophic cascade strength in freshwater ecosystems: a global analysis. *Ecology* 102, e03370 (2021).

Figure 1 – The reader found panel C in this figure to be confusing. The trophic level spread could almost be misinterpreted as the horizontal axis of trophic spread (i.e., habitat coupling) instead of the intended vertical variance around the mean trophic level. It may be helpful for the reader if the authors were to change their language here or to link panel C to panel B more explicitly by having boxes cover the mean/variance within panel B. The authors should also change the title of B (ii) to trophically balanced to maintain consistency with how this is being modelled and avoid confusion with balanced harvesting practices (see comment above Ln 79-80).

Answer: We have changed the presentation in Fig. 1 following these suggestions. First, we have replaced the term “Trophic Level Spread” by “Variation in Trophic Level” and clarified in the figure legend that this is referring to the variation in trophic levels across the three harvested species targeted by fishery fleets. Second, we have changed the panel label of B from *Balanced* to *Trophically Balanced*. We appreciate these constructive improvements of Fig. 1.

Figure 2 – These bar charts are a little hard to look at as they are quite small. The authors may consider stacking the ecological indicators above the economic indicators and moving the legend to the top so that the reader is able to more clearly see the results.

Answer: We followed these suggestions and increased the size and readability of Fig. 2 as described in this point. We are thankful for the detailed assessment of our figures, which has helped to improve the presentation of our results.

Figure 3 & 4 – Here in panels A, the authors have shown how relative species persistence changes with mean and sd of harvesting. As the pristine runs have up to 25% difference in the number of species and the dynamics of these webs may thus be different, it may be worthwhile to include the raw species persistence in the supplementary material.

Answer: In the main part of our manuscript, we show species persistence as the number of persistent species after the simulation of fishery relative to the number of species prior to the simulations of fishery. The latter represents the state after the pristine run, where the food webs have been simulated to remove unstable species that become extinct irrespective of fishery effects.

In this revision, we have followed the suggestion above and added a figure illustrating food-web persistence as the number of persistent species after the simulation with fisheries relative to the number of species prior to the pristine run (Supplementary Fig. SI 8). While similar conclusions can be drawn, this calculation of species persistence is blurred by the effects of random extinctions prior to the simulation of fisheries.

Ln 329 – “the” is repeated

Answer: Corrected.

References

Garcia, S. M., Kolding, J., Rice, J., Rochet, M.-J., Zhou, S., Arimoto, T., Beyer, J. E., Borges, L., Bundy, A., Dunn, D., Fulton, E. A., Hall, M., Heino, M., Law, R., Makino, M., Rijnsdorp, A. D., Simard, F., & Smith, A. D. M. (2012). Reconsidering the Consequences of Selective Fisheries. *Science*, 335(6072), 1045–1047. <https://doi.org/10.1126/science.1214594>

Jacobsen, N. S., Gislason, H., & Andersen, K. H. (2014). The consequences of balanced harvesting of fish communities. *Proceedings of the Royal Society B: Biological Sciences*, 281(1775), 20132701. <https://doi.org/10.1098/rspb.2013.2701>

Zhou, S., Smith, A. D. M., Punt, A. E., Richardson, A. J., Gibbs, M., Fulton, E. A., Pascoe, S., Bulman, C., Bayliss, P., & Sainsbury, K. (2010). Ecosystem-based fisheries management requires a change to the selective fishing philosophy. *Proceedings of the National Academy of Sciences*, 107(21), 9485–9489. <https://doi.org/10.1073/pnas.0912771107>

Reviewer #2 (Remarks to the Author):

Review of "Maintaining ecological stability for the sustainable economic yield 2 of multispecies fisheries in complex food webs"

The authors use a network approach to understanding the system-wide impacts of multi-species fisheries under a variety of fishing scenarios (different target species) and find that due to complex interactions between ecological and economic dynamics, and the cascading effects of harvesting throughout food webs, selectively targeting similar intermediate-trophic-level species can provide a win-win scenario that benefits fisheries sustainability and revenue along with ecological resilience. This is an important result, and I think more notably, the whole food web approach of this study is an important contribution to the field. The authors' approach specifically addresses scenarios where multiple independently-operated fisheries (i.e., each with their own market dynamics) harvest from the same food web. This is an interesting and novel approach that nicely falls in between classical single species fisheries approaches and those that consider harvesting impacts of size- or trophic guild-focused fisheries that often ignore individual species.

Answer: We are very thankful for this positive evaluation of our food-web approach to understanding the consequences of multiple fisheries.

While I think this type of approach is needed in the literature, is timely, and the results here deserve to be communicated to a broad audience, I do feel as though I was left wanting a little more information on the intersection between the fisheries and the ecological impacts, and I found that the results may be somewhat hard to decipher for a general audience. I do however think that this could be addressed by the authors relatively easily and therefore the manuscript certainly has potential to be a high-impact paper. I will elaborate more on these thoughts below.

Answer: We highly appreciate these constructive points that have helped us to strengthen the manuscript and make it more accessible for a general audience.

General comments:

In general, while the results were nicely described in the text, I felt that I was left wanting a little bit more information on the ecological effects of some of the results. One way to address this may be to create a more succinct summary figure of the key results (even if just for the Similar scenario focused on so heavily here) with clear signposting highlighting what is going on. One initial thought I had related to this, and perhaps an example of how to summarize part of the results, is that I appreciated SI 9 showing the biomass structure of the unexploited community, but why not contrast this with the change in biomass structure under a few key scenarios? In my opinion this alone would go a long way in communicating the ecological effects of the fisheries at scale of the whole food web. Another suggestion would be to create a summary figure better highlighting the mean trends in key situations (i.e., the Similar scenarios) to describe important results such as that discussed in Lines 197-203.

Answer: This is an excellent point that we have overseen while assembling the results for the original version of the manuscript. We have followed the suggestion above and added a figure showing how the biomass changes depend on the trophic

levels of the species across the fishery scenarios (new Fig. 3). These new results show that “The fishery scenarios *Prices* and *Distance* primarily lead to biomass reductions in upper trophic level species. This aligns with their respective targeting strategies: *Prices* focuses on high-trophic-level species that yield higher market prices, while *Distance* aims to harvest species across both high and low trophic levels to maximize the number of links between target species. In contrast, the other scenarios predominantly impact the biomass of mid-trophic-level species (new Fig. 3, lines 104-110).” We highly appreciate this addition to our manuscript that has helped to strengthen the communication of our results.

In a similar sentiment, I found myself wanting to understand some of the results more deeply and did not feel that I was provided the information to do so. In terms of the food web effects of harvesting, for example, I am curious how sensitive the sustainability of intermediate trophic level fisheries – and the resulting food web resilience – is to the strength of interactions (and therefore structure) in the underlying food web? Presumably, weaker average interaction strengths and a less top-heavy food web would be more susceptible to fishing selectively at both top and intermediate trophic levels (since there is less energy available to top predators). But is this a matter of the general results holding consistently, or would a switch to lower trophic level fisheries be deemed more sustainable under this scenario? Sensitivity/robustness to factors such as this were not explored very thoroughly, in my opinion, and yet it is these fundamental patterns that are needed for a generalizable fisheries theory. A question remains: are these results robust in spite of the food web structure or because of it?

Answer: During our discussion, we recognized that this is a particularly important point. We agree that systematic changes in food-web structure (which populations are connected by feeding interactions) or interaction strengths (the energy fluxes through these links) could substantially influence our results. As suggested above, stronger energy fluxes (or higher interaction strengths) could reduce the vulnerability of high-trophic-level species to disruptions in energy channels from lower trophic levels. Conversely, weaker fluxes would likely diminish the strength of trophic cascades, thereby reducing the sensitivity of low-trophic-level species to changes at upper trophic levels. While such scenarios are plausible under certain network and interaction-strength configurations, we would like to emphasize that we constrained variation in food-web structures and interaction strengths to remain within the empirically established boundaries provided by meta-analyses. Within these empirically realistic boundaries, we randomly varied model parameters to account for natural variation across communities. Despite this variation, our results consistently highlight the critical role of low-trophic-level species in sustaining energy flows and the potential risk of triggering trophic cascades by harvesting high-trophic-level species. These findings underscore the ecological suitability of targeting intermediate trophic levels for fishing. We have clarified and expanded on this point in the revised discussion (lines 228–240) and are grateful to the reviewer for the thoughtful and thorough consideration of our approach.

It would be useful to contrast your results with other proposed harvest strategies in the literature, along with real world scenarios. As an example (and certainly not the only option),

“balanced” fisheries have been suggested as a way to preserve community structure. You also indirectly come close to this rationale (though through a different mechanism) when discussing the impacts of selecting for similar intermediate TP species and, in a way, balancing competitive interactions. But balanced fisheries have been met with a lot of criticism due to the unlikely feasibility of perfectly matching catches to production of all species in a food web (which is a dynamic property). Do any of your results speak to this or other strategies suggested/patterns seen in the literature (or the real world)?

Answer: We now better relate our results to fishing strategies that have been proposed in the literature and that are applied in real-world fisheries, and expanded the section “Applications to the sustainable management of multiple fisheries” accordingly. The most promising fishing strategy in our model analysis does not strictly follow the balanced harvesting approach, but rather a focus on similar species in the food web seems to be preferred. Yet, the similarity with the balanced harvesting approach is that selective fisheries on the top of the food web should be avoided. We also discuss how our results may inform the increasing debate about initiating fisheries on mesopelagic stocks: Fishing on this kind of species at the lower trophic levels does not seem to be a sustainable strategy. We have included this in the discussion section (lines 261-269).

Similarly, can you give examples of where different fisheries approaches may typically be seen, and if so can you make any inferences about their sustainability?

Answer: We have followed this suggestion and introduced the scenarios with some real-world examples in the discussion (lines 187-197). Adding information on the sustainability of these real-world fisheries is difficult as they often overlap and occur simultaneously.

Next, I will outline some questions I was left asking myself, that perhaps the authors could address through further visualization, additional discussion, or supplemental information:

- How does the equilibrium catch or effort of different fishing scenarios look relative to the community structure? Being able to visualize, for example, different trait distributions (body size, trophic level, productivity, etc.) of the community, and where harvesting occurs relative to this, would be a helpful supplement to Figure 1 and allow for better interpretation of the community-wide harvesting impacts. This could also be extended to a summary of the economic traits of the targeted species.

Answer: We have realized that we have omitted the reasoning behind our choice of trophic level as the main species trait in our analyses. In the revised version of our manuscript, we outline that the three species traits mentioned in the point above, namely body mass, trophic level and productivity, are highly correlated in our models as well as in natural ecosystems. Based on our hypotheses and the definition of the fishery scenarios, we have chosen trophic level as the trait to represent them. We have clarified this point in the revised version of the manuscript (lines 124-131).

- Similar to above, what is the evenness of the catch or effort in the different scenarios?

Answer: In response to this comment, we have added a supplementary figure showing the evenness of the biomass caught across the three fisheries (Supplementary figure S1, lines 110-111).

- Are there trade-offs between resilience and revenue in the Similar outcome, or does this scenario optimize both? (see approx Line 111)

Answer: One of our core findings is that systematically targeting fish resources at intermediate trophic levels optimizes both ecological stability and economic revenue. In the revised manuscript, we have highlighted this result more prominently (lines 165-170). This approach avoids reductions in species persistence and food-web biomass losses (Fig. 5A, B). The higher sustained biomasses of these resilient communities lead to increased biomass catches for the fleets (Fig. 5C) and, consequently, higher sustained revenue (Fig. 5D). Surprisingly, and contrary to general expectations, our results demonstrate that there is not necessarily a trade-off between fishery exploitation revenue and ecological resilience. We believe this is a very encouraging outcome, and we are grateful for the feedback that prompted us to emphasize this key finding more effectively.

Overall, I think the feeling of wanting to understand the drivers of the different outcomes, and therefore the robustness/generality of the conclusions drawn, are reflected in these comments and remaining questions. I do not think it would take a substantial amount of work to address these, but some notable effort into elaborating on these results (and communicating/visualizing them) would greatly improve this paper and explain the importance of these results more broadly.

Answer: We sincerely appreciate the time and thoughtful effort the reviewer has dedicated to assessing our manuscript. We believe the revised version has been significantly improved by incorporating these valuable points and suggestions.

Minor comments:

Table 1 and main text: try and make sure wording is consistent

Answer: We followed this suggestion.

Line 78: is the magnitude of catch (biomass) or the rate of exploitation distributed evenly?

Answer: In the revised version of the manuscript, we have clarified that the choice of target species maximizes their distribution across trophic levels (lines 78-79). We apologize for the unclarity in the original version of our manuscript.

Page 2 in general:

- Clarify in the text that the metrics are relative to an unfished community

Answer: Done (lines 89-90)

- It may be useful to include data on relationships between different fishing strategies. For example, price-driven harvesting and the Similar scenario here with focus on top trophic levels are likely harvesting similar species.

Answer: We have implemented this suggestion by explicitly highlighting the differences between the fishing scenarios when referencing Fig. 4 (lines 134-141).

Line 203: Increase in number of viable fisheries? Should this be decrease?

Answer: Yes, thanks for spotting this error!

Line 327: Distanced scenario – does this unintentionally select for species that are not highly connected within the food web, or ones for which there is little redundancy for species that do interact with them? Could this be a driver of the lack of fisheries sustainability in this scenario? Similar to my comment above about trait distributions, having a better idea of how harvesting is distributed throughout food webs may be very insightful.

Answer: We have clarified that the Distance scenario maximizes the distance in terms of the number of links between the target species and thereby also “minimizes (i) the likelihood that the species are connected and (ii) the redundancy of their roles in the food web” (lines 382-385).

Line 330 and potentially within figures if possible: when referring to the different target trophic groups within the Similar scenario, referred to here as (5), perhaps name these 5a, 5b, etc.

Answer: We followed this suggestion and referred to these subsets of the *Similar* scenario as *Similar low*, *Similar intermediate* and *Similar high*. (lines 387-389 in the methods and throughout the main text of the manuscript)

Missing details in Methods (sorry if I missed any of these!):

- Rationale for values or distribution of important parameters such as γ and R_{opt} may be helpful.

Answer: All parameters in our food-web models are derived from empirical scaling relationships with body mass. This enables model parameterization based on the individual body masses of species, which are sampled from distributions. Supplement SI 13 provides an overview of all model parameters, including their definitions, selected values, and references. In the revised manuscript, we have explicitly emphasized the reference to this supplementary information in the methods section (lines 328-329).

We agree that a concise summary of the empirical scaling relationships employed in our modeling approach would be beneficial. However, given the extensive number of parameters, such an effort extends beyond the scope of this manuscript. In response to this comment, we plan to present these scaling relationships in a more detailed methodological paper in the future.

Regarding the specific question raised, we clarify that R_{opt} represents the optimal consumer-resource body-mass ratio, at which maximum attack rates are achieved.

The parameter gamma defines the width of the Ricker function as a scaling parameter. Both R_{opt} and gamma have been parameterized using values from previous studies (Schneider et al., 2016 and references therein) and calibrated to empirical fits from natural marine food webs (Li et al., 2023).

Schneider, F. D., Brose, U., Rall, B. C. & Guill, C. Animal diversity and ecosystem functioning in dynamic food webs. *Nat. Commun.* **7**, (2016)

Li, J. *et al.* A size-constrained feeding-niche model distinguishes predation patterns between aquatic and terrestrial food webs. *Ecol. Lett.* **26**, 76–86 (2023).

- Some parameters are not defined in the Methods. Examples include $b_{0_herbivore}$ and $b_{0_carnivore}$, β

Answer: We realized that a part of a sentence was missing in the previous version of our manuscript. In the revised version, we have clarified that the parameters b_0 and β represent the intercepts and exponents, respectively, of the allometric scaling relationships for the capture coefficients (b_{ij}) with body mass (lines 375-378). We are grateful that this omission was brought to our attention.

- Please explain how you are defining productivity and if it can change with fishing pressure and changes in food web structure. If your measure is indeed dynamic (and potentially perpetually changing if dynamics are oscillatory), how do you account for this?

Answer: This part was indeed missing. We added the following explanations (lines 390-392): “Species productivity was defined as the difference between energy influx and metabolic rate, which correspond to the amount of energy available for biomass increase in the absence of predation. This calculation has been made at the end of the pristine simulations and therefore represents a fixed property of the ecosystem before the fishery disturbances” We highly appreciate that the reviewer has brought this gap to our attention.

- Some explanation and justification on how all of the values for costs, clearance, P_{base} , etc. were chosen would be helpful, as would knowing if these affect the robustness of the results in any way.

Answer: We have added this explanation to the supplement (see SI 13.4).

- Line 390: “capped to run up to 24 hours” – do you mean of computation time? This was not immediately clear to me

Answer: Yes, this interpretation is correct. We have revised the manuscript to state that “Each food web simulation (food web generation, running the pristine and harvest scenarios) was limited to a maximum runtime of 24 hours.” (line 451)

Figure 1: consider numbering/labeling the choice components in Fig 1A (random, productivity, price) as such and have this match the order presented in the caption.

Answer: We followed this suggestion (see revised Fig. 1).

- Additionally, here and in the text where relevant, consider referring to the “Balanced” scenario as “tropically balanced” or some variation to differentiate between this and what is already known as balanced harvesting in the literature.

Answer: We followed this suggestion.